# Persona-PhysioSync AV: Personalized Interaction through Personality and Physiology Monitoring in Autonomous Vehicles

**DOI:** 10.3390/s24061977

**Published:** 2024-03-20

**Authors:** Jonathan Giron, Yaron Sela, Leonid Barenboim, Gail Gilboa-Freedman, Yair Amichai-Hamburger

**Affiliations:** 1Advanced Reality Lab, Sammy Ofer School of Communication, Reichman University, 8 Ha’Universita st., Herzliya 4610101, Israel; jonathan.giron@runi.ac.il; 2The Research Center for Internet Psychology (CIP), Sammy Ofer School of Communication, Reichman University, 8 Ha’Universita st., Herzliya 4610101, Israel; yaron.sela02@post.runi.ac.il; 3Department of Mathematics and Computer Science, Open University of Israel, 1 University Road P.O. Box 808, Raanana 4353701, Israel; leonidb@openu.ac.il; 4Adelson School of Entrepreneurship, Reichman University, 8 Ha’Universita st., Herzliya 4610101, Israel; gail.gilboa@runi.ac.il

**Keywords:** autonomous vehicle, psychophysiology, personality, emotions, trust, engagement

## Abstract

The emergence of autonomous vehicles (AVs) marks a transformative leap in transportation technology. Central to the success of AVs is ensuring user safety, but this endeavor is accompanied by the challenge of establishing trust and acceptance of this novel technology. The traditional “one size fits all” approach to AVs may limit their broader societal, economic, and cultural impact. Here, we introduce the Persona-PhysioSync AV (PPS-AV). It adopts a comprehensive approach by combining personality traits with physiological and emotional indicators to personalize the AV experience to enhance trust and comfort. A significant aspect of the PPS-AV framework is its real-time monitoring of passenger engagement and comfort levels within AVs. It considers a passenger’s personality traits and their interaction with physiological and emotional responses. The framework can alert passengers when their engagement drops to critical levels or when they exhibit low situational awareness, ensuring they regain attentiveness promptly, especially during Take-Over Request (TOR) events. This approach fosters a heightened sense of Human–Vehicle Interaction (HVI), thereby building trust in AV technology. While the PPS-AV framework currently provides a foundational level of state diagnosis, future developments are expected to include interaction protocols that utilize interfaces like haptic alerts, visual cues, and auditory signals. In summary, the PPS-AV framework is a pivotal tool for the future of autonomous transportation. By prioritizing safety, comfort, and trust, it aims to make AVs not just a mode of transport but a personalized and trusted experience for passengers, accelerating the adoption and societal integration of autonomous vehicles.

## 1. Introduction

Autonomous vehicle (AV) systems are experiencing rapid advancements, as highlighted by Chan (2017), showing remarkable technological progress [1]. However, they are still limited in achieving complete automation [2]. Industry research reveals promises of a revolution in traffic safety, mobility, and quality of life. However, the success of AVs depends on their acceptance [3]. Amidst this progress, there remains a gap in our comprehension of driver preferences within decision scenarios that semi-autonomous vehicles (semi-AVs) encounter [4], all except Level 5 Automation. The Society of Automotive Engineers International (SAE) introduced a classification system in 2016 for vehicle automation levels, ranging from Level 1 (driver assistance) to Level 5 (complete driving automation). The levels are defined as follows: (0) Level 0—No Automation: traditional vehicles without any automation; (1) Level 1—Minimal and Optional Automation: includes features like adaptive cruise control; (2) Level 2—Partial Automation: the vehicle’s automated system can control the car, but the driver maintains responsibility and must be ready to intervene if necessary; (3) Level 3—Conditional Automation: the car is primarily operated by the automated system, but the driver must be present to take control in emergencies; (4) Level 4—High Automation: the automated system handles all driving tasks, including parking, although the driver has the option to take control; (5) Level 5—Full Automation: the vehicle is entirely autonomous, functioning like a robotic taxi [5]. As described, despite significant technological advancements, current autonomous vehicles (AVs) up to Level 4 still require active passenger interaction to facilitate safe traveling. The process of AV decision making, shaped by passengers’ inclinations and preferences, underscores Human–Vehicle Interaction (HVI) as a critical domain within the realm of these emerging challenges [6].

The progression towards empowering AVs, particularly those operating in dense, highly populated areas and responsible for transporting individuals safely from one location to the other [7], is contingent on establishing robust trust and social acceptance towards these robotic entities. The level of trust that would result in high acceptance and product penetration has yet to be achieved [8]. This aspect is of paramount importance for the AV industry, as low levels of trust translate to decreased demand, subsequently leading to higher production costs. Conversely, as consumer trust in new technology grows, demand is expected to increase, driving production levels up and, ideally, reducing costs. The paradox within this context arises from the dual nature of the intent to incorporate autonomous driving agents into society. While there is a drive to design and incorporate autonomous machines and vehicles into daily life, over-reliance and autonomous control have been linked to accidents [9]. One of the central and primary goals of autonomous driving is to provide its passengers with the leisure to participate in other activities such as work, watching a video, or simply enjoying the ride without the stress and demands of driving [10]. Consequently, and unwantedly, the human component within this interaction assumes a vital role, especially in Level 3 [5], diminishing the potential of AVs to provide this leisure. Therefore, we learn that the autonomous component cannot rely only on technological aid, but human involvement must be maintained.

The experience within an autonomous vehicle should enable passengers and drivers to engage in non-driving-related tasks (NDRTs) while keeping relevant engagement with the journey and limiting motion sickness and maintaining high comfort [11]. Presently, numerous sensors are being integrated into the cabins of autonomous vehicles (AVs) to facilitate advanced interaction, enabling the understanding of driver and/or passenger actions, emotions, and personal preferences. This integration aims to provide adequate functionalities and services for a safe and enjoyable drive [6]. Among these interfaces, some focus on detecting attention, emotion recognition, drowsiness, and mental workload [6]. These interfaces prioritize evaluating and ensuring the proper state of the driver/passenger to maximize safety levels during the drive using driving assistance systems [6].

A critical aspect of the interaction between the autonomous vehicle and the passenger/driver are Take-Over Requests (TORs); these are crucial and must be acted upon quickly [12]. The TOR transition necessitates the passive passenger’s shift into an active driver role, which poses a recurring challenge. During this transition, the driver, who has been in a passive role, is asked by a driving assistance system to take control of the vehicle navigation system and wheel. One of the risks that they may experience is a loss of situational awareness (SA) [13] required to assume the active role swiftly [14,15]. Nevertheless, as outlined in Vogelphol et al. (2018), individuals who are highly distracted need more time to restore their situational awareness and respond appropriately to the TOR [16]. The TOR represents one of the most critical interaction challenges between an autonomous vehicle and its driver/passenger. This interaction is paramount for ensuring passenger safety and, in turn, bolstering trust in the system. Therefore, confirming passenger engagement is essential to guarantee appropriate reaction times when necessary, such as TOR [17,18].

Du, Yang, and Zhou (2020) presented evidence indicating that cognitive workload, emotional states, attention levels, and engagement play pivotal roles in the TOR process; as such, it is imperative to monitor these factors [12]. In the paper by Melcher et al. (2015), the authors present the challenge of adequately designing the TOR strategies. These include different strategies for the initiation of the TOR as well as integration with brake jerk stimuli and an audio one [19]. Although they did not find a significant difference between their test groups, they report on the importance of integrating the TOR request as part of the NDRTs (in their case, a mobile application in which they are engaged) which results in higher perceived safety of the system [19]. In a study conducted by Zue et al. (2023), TOR design was assessed while passengers were reading or watching videos in NDRTs. Their findings indicate that incorporating tactile-auditory or tactile-visual cues with 7-s and 9-s TOR Lead Times produced positive outcomes in reading scenarios, while tactile cues showed effectiveness in video watching scenarios [20]. As presented, the process of reacquiring control of a vehicle following automation is of critical importance. This process necessitates that the driver maintain awareness of the specific emergency situation, process information conveyed by the system, and discern and comprehend the prevailing traffic conditions. Several factors contribute to the complexity of this task, encompassing objective factors such as traffic conditions, road conditions, and the control interface, as well as subjective factors including NDRTs, age, trust, urgency, the human–machine interface (HMI) employed, and SA [21]. Current solutions include the evaluation of different interaction paradigms that aim to recruit passenger engagement in low response times [6] and the incorporation of human-like behaviors into the operational parameters of the vehicles [22]. The project Mediator (European Union’s Horizon 2020 research and innovation programme) was designed to develop a system that can determine the most suitable choice between using a car and having a driver based on who is best suited for driving at any given moment. In our current work, we suggest expanding on current perspectives and unlocking a deeper understanding of users’ state and engagement levels by incorporating additional subjective factors into the measurement process such as their personality traits [23] and psychophysiological responses [24,25]. Our goal is to introduce a novel interface to aid in this aspect for AVs up to Level 4, encompassing those involving TOR interaction. The integration of subjective measures related to personality and physiological traits may evoke apprehensions regarding data privacy [26]. To mitigate this, it is imperative that the system guarantees full privacy and encryption in data collection. By offering complete assurance of data privacy, the system can leverage the potential benefits of data collection while fostering a balanced interaction with the AV, thereby nurturing trust and bolstering acceptance.

## 2. Trust in the Safety of the Autonomous Car

Trust in and acceptance of technology are significantly influenced by an individual’s personality [23], disposition [27], and physiological tendencies [28]. A fundamental aspect of engaging in semi-AV driving involves transitioning from an active driver role to a passive passenger role and the other way around. This distinction gives rise to substantial concerns regarding the adoption of AVs and their market expansion [29]. Although highly advanced and capable, the autonomous vehicle, at certain times, would choose to relinquish control and activate a TOR strategy in order to navigate it over the current obstacle or path with the help of the user. These reservations manifest themselves in reduced levels of trust and societal acceptance of AVs, thereby limiting the potential for the widespread adoption of such systems. Ganesh (2020) further highlights the necessity of incorporating human controls into the vehicle’s operational framework as a complementary mechanism to enhance agency and elevate safety, specifically in terms of information exchange between the vehicle, its occupants, and drivers [30]. Effective communication is vital, ideally through interactions that engage users without overtaxing their cognitive and emotional resources based on a human-centered design approach [31] Interaction mechanisms vary, ranging from voice commands [32] and touch-based inputs to gestures [6]. These interactions can be explicit, where the user is fully aware of the interaction, or implicit, occurring subconsciously. This assumption was displayed in the work by Nakade et al. (2023), who use haptic co-driving to show such concepts [9]. Applying this approach can lead to a smoother, more comfortable experience for passengers and fewer issues with the flawed operation of TORs.

It is suggested that the interaction with autonomous vehicles (AVs) might reflect the subtle, implicit interactions found in human–human collaborations during joint physical tasks or be akin to the way individuals use glasses for reading, where there is an underlying sense of merging or collaboration with the vehicle. Researchers such as Fuchs, Ganesh, and Nakade have explored these concepts [9,30,33]. The development and application of suitable implicit interfaces could potentially foster these sensations. By integrating human elements into the interaction systems between humans and vehicles, particularly through the ongoing monitoring of physiological responses, movement, eye tracking, and more, trust and acceptance are likely to increase. To cultivate trust, AVs should not solely rely on their sophisticated visual capabilities for navigation. They should also take into account the intentions of the passengers [34], their personalities [23,35,36,37], and levels of engagement [18]. In Nakade et al. (2023), a driver-centric automation strategy was introduced with the aim of achieving collaborative steering through haptic co-driving [9].

Our interaction framework involves two distinct stages, each tailored to different aspects of the autonomous vehicle (AV) experience. Firstly, our framework prioritizes safety during TOR transitions. This phase is crucial as it marks the shift from autonomous driving to human control. To ensure the safety of the drive, the AV employs comprehensive safety protocols, basing the operation on real-time monitoring systems provided by the framework. This includes rigorous testing and validation of the AV’s sensor fusion algorithms and decision making processes to accurately assess the driving environment and respond to potential hazards effectively. Additionally, clear communication channels are established to alert passengers about the transition and provide reassurance regarding the safety measures in place. Secondly, the framework seamlessly transitions into a mode focused on enhancing user comfort throughout the drive. This involves optimizing the AV’s driving behavior to prioritize passenger comfort, such as smooth acceleration, braking, and steering. Furthermore, the framework incorporates personalized settings and preferences to tailor the driving experience to individual passengers, including options for climate control, entertainment, and seating adjustments. By prioritizing passenger comfort, the framework aims to enhance the overall user experience and promote trust in and acceptance of autonomous driving technology. Based on the suggested implicit interaction framework, in this paper, we will propose a set of measures that can be incorporated into autonomous vehicles. These sensors, in conjunction with personality trait measurement, can offer an appropriate interface to monitor passenger engagement levels. This can help in adapting driving strategies to achieve the successful transportation of riders as well as TOR transitions. By implementing this method, the AV will be able to interpret both explicit (verbal, touch movement, etc.) and implicit signals from users. This will lead to enhanced HVI that is expected to bolster user acceptance and trust in AV technology. The AVs will adjust their driving and navigation based on these signals to ensure that users are engaged and comfortable. For example, the AV may alert users if they are drifting away or not paying attention, or it may adapt its driving patterns to reduce discomfort for passengers. The operation should be unobtrusive yet engaging, providing necessary control to avert potential disasters. We suggest the use of an array of sensors that will provide an implicit interface between the vehicle and the passenger/driver, resulting in a more natural way of interaction, one that induces a cooperative interaction within the HVI. We propose a conceptual framework referred to as the Persona-PhysioSync AV (PPS-AV) in Figure 1, designed to improve how users interact with semi-AVs. This PPS-AV framework is based on aligning the interaction with the user’s personality traits, as described by Amichai-Hamburger et al. (2022) [23]. Additionally, the PPS-AV framework considers users’ current psychophysiological and emotional state, indicated by factors like facial expressions. By integrating personality traits with real-time emotional and psychophysiological data that produce the state reaction of the interaction, the PPS-AV framework aims to tailor the interaction style between the user and the vehicle. This tailored approach is intended to enhance the user’s trust in and acceptance of the autonomous vehicle. The current work aims to elucidate the hypothesis that natural cooperative interaction with autonomous vehicles is crucial and might be even more critical than safety measures in elevating trust in and acceptance of the technology.

The PPS-AV framework draws on research by Pascale et al. (2021), which indicates that trust in technology, specifically autonomous vehicles (AVs), can be enhanced through direct experience [38]. Their study reveals that after individuals ride in an autonomous car, they tend to have a more favorable attitude towards this technology, showing increased acceptance and decreased risk perception. This change is particularly significant, as participants noted less false hazard detection during their experience [38]. These findings underscore the importance of human interaction with cars, suggesting that aligning driving patterns with passengers’ expectations and comfort levels could lead to greater trust in and acceptance of AVs.

Tailored interactions and interfaces, which are non-intrusive and require minimal cognitive effort [39] but elevate motivation [40], are particularly vital in the context of self-driving cars. Hartwich et al. (2021) conducted a simulator-based study wherein they assessed the impact of different human–machine interfaces (HMIs) on passenger experiences, with a particular focus on trust [29]. The study involved a comparison between a context-adaptive HMI and a fixed (permanent) HMI. Additionally, the researchers controlled for initial trust levels, distinguishing between low-trust and high-trust groups among their participants. The findings revealed a significant reliance on participants’ initial levels of trust and their perception of trust in various HMIs. Specifically, the low-trust group displayed a preference for the permanent HMI, while the high-trust group exhibited greater comfort with the context-adaptive HMI [29]. These results suggest the potential benefits of tailoring autonomous systems to individual users to enhance trust and overall user experience. Inducing trust in and acceptance of a system relies on the perception of risk and uncertainty [41].

## 3. Sensing Users and Tailoring the Experience of the Autonomous Car

It has been demonstrated that an individual’s personality influences how they are likely to use technology [42,43], specifically AVs [3,23,35,44]. Physiological tendencies were also found to play a role in the interaction between users and technology and how they use it [45] and even more so in driving styles and interaction with AVs [46]. The concept of creating interfaces that are based on personality was developed by Amichai Hamburger (2002) [42]. The link and prediction ability between personality traits and physiological response has been previously studied [45,47,48,49,50], specifically in the AV setting [51]. Engagement with the technology and attention to it were critical to induce adequate interaction [52] and elevate levels of trust in and acceptance of it [53]. Personality measures provide valuable trait classification of users [54], and physiological tendencies and ongoing responses provide the dynamic aspect of user interaction [54]. Incorporating these tendencies can be a key predictor of their level of engagement and attention while interacting with the technology.

We suggest a framework that will be based on the development of a continuous monitoring protocol of passengers’ comfort [55], engagement [56,57], stress [58,59], and fatigue levels [60] through multimodal physiological and emotional parameters [61] adjusted to their personality traits (Figure 1) [62]. By proposing this protocol, we aim to create a new way for passengers and drivers to interact with their AVs. This approach will help design a personalized human–machine interface that gathers essential information about the user’s engagement and comfort. The vehicle can then use these data to adjust its driving style accordingly. This strategy is expected to elevate trust in and acceptance of this technology, as it tailors the driving experience to individual needs and preferences.

### 3.1. Personality

Amichai Hamburger et al. (2020, 2022) suggested that individualizing the autonomous travel experience using personality trait measures can induce the confidence of passengers in the car [23,63]. One of the most influential models about personality is the Big Five, which states five significant factors of personality: (1) openness to experience (inventive/curious vs. consistent/cautious), (2) conscientiousness (efficient/organized vs. easy-going/careless), (3) extroversion (outgoing/energetic vs. solitary/reserved), (4) agreeableness (friendly/compassionate vs. challenging/detached), and (5) neuroticism (sensitive/nervous vs. secure/confident) [64], which were found to be linked to driving styles [23] and crucial in designing intelligent systems and engagement classification [65]. Furthermore, individuals with different needs for control in personality traits respond differently to a given situation. Those in high need of control would feel less comfortable relinquishing control to the car. When it comes to the “locus of control” of individuals, it seems the tendency is to blur the lines and abdicate responsibility to technology. This results in overconfidence in the technology’s ability to keep them safe, which results in slower intervention in autonomous control such as cruise control. However, there are clear signs of system failure [66]. Similarly, their reckless driving pattern results in the belief that their chances of getting into an accident are out of their control [67]. When contemplating sensation seeking traits, it was expected that individuals who are high on the thrill and adventure seeking scale and are high on experience seeking, reported as sensation seekers, drove faster, left less headway between cars, and tended to brake later and harder [3,67].

### 3.2. Physiological Sensing Indices

Physiological measures provide a window to the emotional state of an individual. As indicators of the activation of the autonomous system, these measures provide a glimpse into the dynamic activation of the sympathetic nervous system when at unease or stress and the default activation of the parasympathetic system when at ease [68]. The extraction of engagement and trust from such physiological parameters was explored by Mühl and colleagues (2020) [69]. Using an autonomous vehicle and a simulator, they aimed to test whether trust is reflected in physiological responses. They measured trust through self-reports, skin conductance, and driving scenarios. Participants experienced the same route with both human and autonomous drivers. Results indicated higher trust in human drivers, confirmed by physiological measures like skin conductance. The study also found a preference for human drivers using a defensive driving style [69]. Feasibility has been demonstrated in measuring individual responses using physiological indices, including stress and emotions while driving, in real-life scenarios [69]. Further evidence of the possibility of evaluating users’ emotional responses using physiology was shown in [61,70,71,72] using relevant simulators [73,74]. Tavakoli et al. (2021) presented the HARMONY experimental platform, which can analyze drivers’ states in naturalistic driving studies using multimodal data collection [71]. This includes in-cabin parameters such as light, noise, and ambiance, driver measures such as heart rate, facial, gaze, and pose features, and outside measures based on global positioning system (GPS) and video-based AI [71]. Utilizing physiological sensing technologies such as heart rate, electrodermal activity, pupillometry, and others has proven to be valuable for revealing cognitive states, motion sickness [75], and drowsiness [76] as well as reducing distraction levels while measuring engagement levels while interacting with an AV [6]. Automatic engagement detection was shown using wearable sensors [77]; the classification of engagement while learning using these methods reached up to 85% classification rates. Belle et al. (2011) present a signal processing system aimed at detecting optimal engagement and attention with successful results in distinguishing between attentional states [78]. Katsis et al. (2008) performed an evaluation of car-racing drivers’ emotional states using facial electromyograms, electrocardiograms, respiration, and electrodermal activity, reaching 75% accuracy rates [79]. As presented, an array of measurements can provide the required interface and support the suggested implicit interface.

### 3.3. Validated Psychophysiological Sensors

The PPS-AV framework should be equipped with advanced sensors designed to track and assess passenger engagement and comfort inside the AV. We advocate for a comprehensive multimodal approach, which simultaneously analyzes multiple measures. This strategy is based on the following key indices.

#### 3.3.1. Electrodermal Activity (EDA)

Electrodermal activity (EDA) is a biosignal that reflects an individual’s level of arousal by measuring the electrical properties such as conductance variance induced by sweat in their hands during an experience [80]. This method has been extensively researched and validated in numerous studies, including those specifically evaluating arousal in autonomous vehicles (AVs) [81]. Su and jia (2022) used EDA features to detect the comfort levels of passengers in an AV [82]. In a study by Zheng et al. (2015), various biosignals were used to quantify drivers’ stress responses in autonomous trucks [83]. The researchers utilized palmar perspirations, EDA, masseter electromyography, and teeth clenching to assess the stress responses of drivers in autonomous trucks, establishing a correlation between the inter-vehicular distance (distance between trucks) and the corresponding psychophysiological reactions. This study was contextualized within an operational framework to minimize the spatial gap between trucks to reduce aerodynamic drag caused by wind resistance. The primary focus of the investigation was to evaluate the impact of this proximity reduction strategy on the well-being of drivers and passengers within these vehicles. They concluded that reported stress significantly increases as the gap between trucks decreases [83].

#### 3.3.2. Electroencephalography (EEG)

EEG is a technique for recording the brain’s surface layer’s macroscopic activity through electrical activity on the scalp using dry contact electrodes or conductive gel. It was found to be a useful tool in driver distraction research, showing promise in detecting driver fatigue or sleepiness [84,85]. EEG is differentiated into beta, alpha, delta, and theta bands based on the frequency range. Theta waves mark sleepiness onset, while delta activity indicates a sleep state [86]. Some examples of suggested in-vehicle interaction are the recording of EEG for measuring fatigue and sleepiness [84,86,87]. Furthermore, the use EEG in a work scenario has proved helpful as well [88].

#### 3.3.3. Electrooculography (EOG)

The use of optical sensors to track eye movement and blinking patterns was validated to provide insights into cognitive alertness in drivers [89]. Rapid eye movements signify alertness, while drowsiness leads to slower movements and prolonged blink rates. Using EOG was shown to provide up to 80% effectiveness in detecting driver drowsiness [90]. Other work has shown that EOG can be used to predict cognitive alertness and drowsiness [91].

#### 3.3.4. Electromyography (EMG)

The recording of EMG involves an electrical sensor that records muscle electrical activity. Studies such as that by Katsis et al. (2008) show that driver distraction is marked by reduced EMG signal amplitude and frequency, making it a useful tool for assessing alertness levels [79]. The work by Hussain et al. (2023) expresses the use of EMG for intended action prediction as an addition to the use of cameras and radar- and lidar-based advanced driver assistance systems in AVs [92]. Additionally, the research showcases the effectiveness of the suggested concept through experiments aimed at categorizing online and offline EMG data in real-world environments [92]. Demonstrating the assessment of a driver’s capability to respond to TORs using EMG measurements adequately was exemplified by [93]. The SafeDriving system by Fan et al. (2022) employs EMG sensors and deep learning to identify real-time abnormal driving behaviors [94]. The system utilizes a wearable EMG sensor on the driver’s forearm to gather data related to five predefined abnormal driving actions, each of which is subsequently labeled [94].

#### 3.3.5. Electrocardiography (ECG)

ECG monitoring involves a sensor that records heart activity and rate, offering a more straightforward way to detect drivers’ alertness [95]. It can also indicate a driver’s mood, with high heart rates suggesting excitement or anger and normal rates indicating calmness. ECG was effective in identifying alertness [95] and stress [73]. In the work described by Healey and Picard (2005), driver stress was extracted from heart rate measures known as heart rate variability (HRV), a measure that represented the ratio of sympathetic activation compared to parasympathetic activation [81]. Dillen et al. (2021) present compelling evidence of the differences in responses elicited by various autonomous driving styles [96]. Their paper describes ongoing physiological measurements (EDA and heart rate) recorded while the autonomous vehicle took different driving strategies, including acceleration thresholds or lane change behavior. Their results show that the physiological responses were proportionally related to the magnitude of the acceleration and jerk [96]. The AUTOMOTIVE case study by Esteves et al. (2021) focused on using signal processing and machine learning methods to detect drowsiness in individual drivers within AVs, utilizing immersive driving simulators [97]. They employed ECG and facial data to continuously develop personalized drowsiness models, enhancing monitoring efficiency [97].

#### 3.3.6. Body Temperature

Evaluating the comfort levels of drivers and passengers can involve assessing their body temperature, which can be used to fine-tune the car’s internal climate settings to suit individual preferences. Several wearable systems are available to measure body temperature, with some advanced models offering wireless data transmission to smartphones or in-car electronic devices [98,99,100]. Gwak et al. (2015, 2019) present evidence that changes in indoor ambient temperature can enhance both thermal comfort and occupants’ arousal levels [101,102]. Furthermore, as described by Sunagawa et al. (2023), during brief driving sessions, the individual’s thermal surroundings may exert a more significant impact on the progression of drowsiness than the duration of driving [103].

Variations in body temperature can be indicative of stress, anxiety, or discomfort, all of which can significantly impair cognitive functions and decision making abilities [104]. In the context of autonomous vehicles, where the interaction between human and machine is nuanced and complex, recognizing and responding to such physiological cues becomes essential [105]. For instance, an increase in body temperature could signal stress or anxiety, prompting the vehicle’s systems to initiate calming interventions, such as adjusting the interior lighting, climate, or even suggesting a break in the journey. Furthermore, monitoring body temperature can enhance the personalization of the user experience [106]. By understanding an individual’s typical physiological responses under various conditions, the vehicle can preemptively adjust settings for optimal comfort and performance. This level of personalization not only improves the user’s experience but also fosters a deeper trust and rapport between the user and the autonomous system, crucial for the widespread acceptance and success of these vehicles.

### 3.4. Emotional Measurement

#### Facial Action Coding System (FACS)

Implementing continuous camera input aimed at the driver or passenger’s face allows for the analysis of ongoing emotional states. The effectiveness of using FACS [107] in an autonomous vehicle (AV) environment was confirmed in a study by Meza-Garcia et al. (2021), which demonstrated that simultaneously recording EDA and FACS is highly effective in capturing emotional responses from participants [108]. Additionally, in-cabin recording and emotion extraction from drivers’ expressions were significantly validated in the work by Liu et al. (2021). Their study describes the concept of an “empathic car”, which is capable of reading and responding to users’ emotions during the driving experience [109]. In the study by Beggiato and Rauh and Krems (2020), facial expressions were used to indicate discomfort while participating in automated driving; they concluded that facial action unit analysis can contribute to the automatic detection of discomfort while riding in AVs [110].

Evaluating experiences based solely on a single signal measurement often falls short of accurately representing an individual’s actual state. By integrating a range of signals, as described, including those related to personality traits, and applying multimodal analysis techniques [111], a deeper and more comprehensive understanding can be achieved. This approach is particularly beneficial when subjective reports are not fully available or influenced by social apprehensions.

### 3.5. Personality and Physiological Indices

Personality traits and their interactions with physiological responses have been previously studied. Stemmler and Wacker (2010) describe the personality–physiology relationships for central and peripheral nervous system activity measures. They suggest an interactionist conceptualization of traits as dispositions of the situational context, and its subjective representation by the participants moderated this relationship. They conclude by outlining the implications of the interactionistic approach to passengers’ physiological personality research [50]. Evin et al. (2022) describe a machine learning process that could predict drivers’ personality trait levels using EDA and driving behaviors. Their study evaluated risky urban situation behaviors simultaneously with EDA activity. The machine learning protocols employed in the study resulted in 0.968 to 0.974 prediction levels with better detection of neuroticism, extroversion, and conscientiousness [48]. In a study by Childs, White, and De Wit (2014), individuals with high negative emotionality exhibited considerably more significant emotional distress and lower blood pressure responses to the Trier Social Stress Test [47]. Individuals with high agentic positive emotionality exhibited prolonged heart rate responses to stress. In contrast, those with high communal positive emotionality exhibited smaller cortisol and machine learning responses. Shui et al. (2023) report on the ability to predict Big Five personality traits using physiological signals. In their experiment, they recorded HR-based features in five situations averaged across ten days [49]. This evaluation revealed correlations between HR features of 0.32 and 0.26 for the dimensions of openness and extroversion, with the prediction correlation trending significance for conscientiousness and neuroticism. Their findings demonstrate the link between personality and daily heart rate measures using state-of-the-art commercial devices [49]. Physiology and driver characteristics were found valid for the classification of sleep deprivation and cell phone use. The combination of all three features (physiology, personal characteristics including personality measures, and vehicle kinematics) results in the highest classification accuracy [51]. As illustrated, the strong interconnection between personality and physiological responses underscores the necessity of considering both aspects in tandem. For instance, an individual with high scores in thrill seeking traits will exhibit distinct physiological responses compared to someone who scores low on the same measures. Personality traits constitute the stable component of evaluation, while physiological indices contribute to the dynamic elements of measurements. Together, they provide a comprehensive account of the entire experience. Hence, personalization is crucial in fostering a robust interaction between humans and vehicles.

## 4. Conclusions

The introduction of AVs is a significant technological breakthrough with the potential to transform the transportation sector. User safety is the cornerstone of any mobility technology, including AVs, making it the top priority. While ensuring safety, a secondary goal for AVs is to offer a comfortable experience that enables users to engage in non-driving-related tasks (NDRTs), as outlined in the AV mandate. Despite advancements in safety features, trust in and acceptance of AV technology remain relatively low, which could impede its broader adoption, resulting in low demand and higher production costs. The widespread adoption of AVs may be hindered by the traditional “one size fits all” approach, which could limit their social, economic, and cultural impact. A key challenge identified in the literature is the low level of trust in and acceptance of AV technology. Research indicates the feasibility of measuring users’ sensations and emotional responses during autonomous drives. As highlighted in the work of Hamburger et al., variations in personality traits are crucial in understanding and predicting the appropriate driving style for each passenger (2022) [23]. Additionally, a growing body of research is exploring the link between understanding individual personality traits and their psychophysiological responses. Enhancing user experience and interaction design is critical to building trust in AV technology.

The main paradox, as discussed above, revolves around the seamless transition between autonomous driving and human driver control. Autonomous vehicles (AVs) aim to offer passengers the opportunity to engage in non-driving-related tasks (NDRTs). However, until Level 5 autonomy is achieved, where complete automation is possible [5], fully disengaging passengers poses significant risks. This means that while participating in NDRTs, passengers must maintain a certain level of engagement to respond effectively to TOR events. The suggested PPS-AV framework is designed to resolve this paradox by providing passengers with a dependable monitoring protocol. This protocol allows passengers to disengage to a reasonable extent while ensuring they are promptly alerted if their disengagement becomes excessive. Furthermore, to the best of our knowledge, the integration of personality traits with physiological and emotional indices to classify passengers’ engagement and comfort in autonomous vehicles has not yet been established. This gap highlights the novelty of our proposed framework, as it pioneers linking these elements to enhance the understanding and personalization of passenger experiences in autonomous driving contexts.

For effective interaction with AVs, innovative interfaces are required. These interfaces should monitor personality traits, which affect users’ experiences with AVs, and capture both explicit responses (like speech or behavior) and implicit reactions (such as psychophysiological and emotional indices) that influence the overall sensation during the experience. Providing AVs with the right interface can foster a heightened sense of Human–Vehicle Interaction (HVI), perceived by users as collaboration, thereby fostering trust in the technology. In this paper, we propose a new framework for designing an implicit interface with AVs. This interface aims to monitor passengers’ states to tailor driving styles to individual preferences and track user engagement. It is designed to alert users when they reach critical levels of disengagement and low situational awareness in which they may not respond adequately to TORs, enhancing the overall experience with AVs.

The PPS-AV framework places its emphasis on the initial assessment of passengers’ engagement and comfort levels in autonomous vehicles (AVs). We believe that monitoring individuals’ ongoing states can subsequently help identify conditions unsuitable for vehicle operation during TOR events. Additionally, it can recommend adjustments to driving patterns in response to passengers’ discomfort alerts. Personality traits form the enduring aspect of assessment, while physiological indices add the dynamic facets to measurements. When combined, they offer a holistic depiction of the overall experience. For example, the AV will notify the passenger using naturalistic notifications (haptic, visual, or auditory) when they are too engaged in the NDRTs to pay attention to the road and regain the necessary situational awareness. Without this assessment, identification and intervention would not be possible. The PPS-AV system serves as the foundational level of interaction, paving the way for advancements in Human–Vehicle Interaction within AVs. The next levels will include interaction protocols based on interfaces such as haptic alerts and visual and auditory cues, all based on the concept of emotional haptics [112]. This is for the purpose of circumventing the low engagement of passengers without interrupting their NDRTs.

Another potential application of the system could involve assessing a passenger’s initial condition to determine their eligibility to drive, ensuring they can handle the necessary TOR activation in the required response time to avoid crashing and aligning with their personality traits and inclinations.

A crucial concern demanding attention is the integration of sensors in the automotive sector, which presents two significant challenges: precision and cost. The precision of measurements profoundly affects safety, while cost considerations are pivotal in fully realizing the products’ potential. Therefore, it is imperative to address these challenges during the system’s development. Future research and development efforts should place significant emphasis on resolving these issues, as their successful resolution is essential for advancing the automotive industry.

## 5. Implications

The implications of the personality–physiological interface (PPS-AV) framework and the broader concepts discussed in the paper for the field of autonomous vehicles (AVs) are significant:**Enhanced User Experience**: The PPS-AV framework and the focus on monitoring and accommodating individual passenger preferences and comfort levels can greatly enhance the overall user experience. This could lead to increased user satisfaction and greater adoption of AV technology.**Safety**: By monitoring passengers’ engagement and providing alerts when necessary, the framework contributes to passenger safety. It ensures that passengers remain aware and capable of responding to TOR events promptly, reducing the risk of accidents.**Trust Building**: AVs often face trust and acceptance challenges. The ability to tailor the driving style and level of engagement to individual passengers’ preferences and personality traits can build trust and comfort with the technology, potentially accelerating its adoption.**Personalization**: AVs equipped with the PPS-AV framework can offer a more personalized travel experience. This personalization extends beyond just driving style and engagement levels to include entertainment, climate control, and other aspects of the passenger’s journey.**Human–Vehicle Interaction** (HVI): The development of innovative interfaces that consider personality traits, explicit and implicit responses, and emotional indices can foster a sense of collaboration between passengers and AVs. This HVI can make the interaction with AVs more intuitive and user-friendly.**Research and Development**: The proposed framework highlights the need for ongoing research and development in the AV field. It encourages the integration of psychophysiological and emotional data into AV systems, which can lead to more advanced and capable AV technologies.**Safety Regulations**: As AV technology evolves, the introduction of frameworks like PPS-AV may lead to establishing safety regulations and standards that focus on monitoring passenger states and engagement.**Market Differentiation**: Companies implementing such personalized and safety-enhancing technologies may gain a competitive edge in the AV market, attracting customers who prioritize safety, comfort, and a personalized experience.**Data Privacy and Security**: The collection of psychophysiological and emotional data raises concerns about data privacy and security. Implications for data protection and secure handling of sensitive passenger information must be addressed in the development and implementation of such systems.

The PPS-AV framework and related concepts have the potential to revolutionize the AV industry by addressing some of its key challenges and significantly improving the way passengers interact with and trust autonomous vehicles. These implications can drive advancements in technology, regulations, and user acceptance in the AV field.

In summary, our interaction framework is designed to address both the safety and comfort aspects of the AV journey. By effectively managing the TOR transition and prioritizing passenger comfort throughout the drive, we aim to deliver a seamless and enjoyable autonomous driving experience for all passengers.

## Figures and Tables

**Figure 1 sensors-24-01977-f001:**
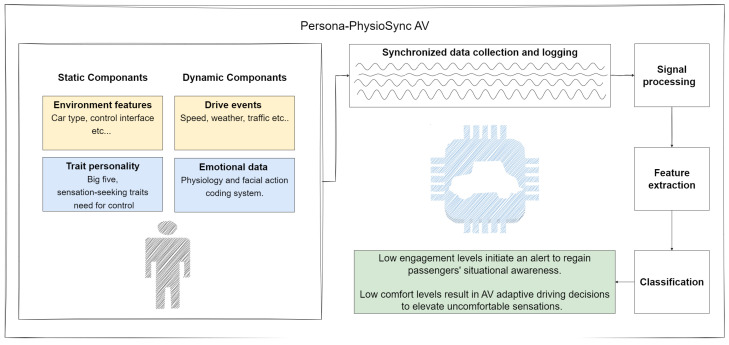
The interface diagram of the PPS-AV illustrates the initial login process for static components like environmental features and trait personality measures. As the user experience unfolds, dynamic components, including psychophysiological and emotional responses, as well as driving events, are logged. These dynamic components encompass heart rate, skin conductance, facial expressions, emotional state, and specific driving events like sudden stops, accelerations, or changes in traffic conditions. The system continuously monitors, records, and synchronizes these data, processes them, and extracts key features. These features are then classified and used to inform the autonomous vehicle (AV), enabling it to respond dynamically to the user’s current state and the driving context. Responses include alerting the user in the case of a loss of situational awareness, adapting the vehicle’s driving style to align with the user’s psychophysiological and emotional conditions, and reacting appropriately to the driving events occurring in real time.

## Data Availability

Data are contained within the article.

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
