# Peer review of "Persona-PhysioSync AV: Personalized Interaction through Personality and Physiology Monitoring in Autonomous Vehicles"

_sensors, 2024, doi:10.3390/s24061977_

Round 1

Reviewer 1 Report

Comments and Suggestions for Authors

Authors are commended on a well-organized paper covering a relevant topic for today's automobile industry. The paper flows well and is easy to follow, and overall, is well thought out. There are several referencing and grammatical errors, as seen in the comments concerning English language, but outside of that, there are only a few points for suggested improvements within the paper before moving forward with publication. 

In the introduction and in the conclusions, the limitations associated with costs of this technology needs to be addressed. Costs are mentioned in the last paragraph before implications, but this discussion needs to be expanded with the understanding if we move forward with the studying of this technology and further application that cost effective strategies can be addressed and implemented. Without addressing the hinderance associated with consumer trust, demand is low, and thus, costs are high. As consumer trust builds with new technology, demand will rise driving production up, and hopefully, costs come down. Another area for further exploration within the paper is focused on trust building in the form of assurance concerning data privacy. This concept is not addressed until the implications section and is only mentioned in the last item listed, number 9. This discussion needs to take place at the beginning within the introduction, and circled back to within the discussion.

Further, for section 3.3.6. Body Temperature, this section needs to be expanded as the discussion seems quite simplistic where the focus is just on comfort on the passenger. The other sections discuss the value of these interactive measures to ensure alertness of the driver. There are several studies looking at temperature within the vehicle and it's influence on driving, specifically alertness of the driver. These studies would justify the need for detecting the ideal temperature for the alertness of the driver, rather than just comfort. 

Finally, since this paper is focused on new technology and a new approach to human-vehicle interaction, references concerning previous work should be relatively new. However, over 1/4 of the references listed, around 27%, are a decade old, if not older. For those references, either remove or include newer references to support these findings. 

Comments on the Quality of English Language

In several lines, as given below, the authors reference the authors of other papers within a sentence, but do not include the year of that publication. This is not done with every reference, and thus, to maintain consistency throughout correct the following lines so that the year of the publication is included:

240, 251, 296, 303, and 320.

For several sentences the authors are referenced with the year of the publication, however, the number associated with that reference as given in the reference list is not included. This number should be given, preferably at the end of the sentence in which this is done for most, but not all sentences where the authors are mentioned. Below are the lines where the authors are mentioned, but no reference number is given at the end of the sentence:

74, 168, 204 (25 is given, but then ? is given for the 2020 publication), 240, 267, 303, 314, 320 (no year given either), and 367

In lines 296, 362, and 396, move the number of the reference to the end of the sentence. 

In the following lines, capitalize the first word at the start of the sentence: 267, 340, 374, 439, and 441.

In lines 38, 40, and 267, remove the period before the reference number.

In line 282, add a period before EEG. 

In line 332, remove [105] at the beginning of the sentence, before the word Implementing. 

In lines 11, 60, and 464, TOR is defined, but the authors define it as Takeover Request, Take Over Requests, and Take over requests, respectively. Authors need to be consistent in how they define TOR in both the capitalization and whether they combine or separate the words take and over. Go with what is the most common within the industry and what is utilized most consistently within the research. 

Author Response

Dear Editor,

We thank the reviewers for the thought and time they put into reviewing our manuscript and are grateful for their generous remarks and insightful comments. We have edited the article to address their concerns and, in so doing, have improved it greatly.

We believe that the article is now suitable for publication in Sensors

Dr. Jonthan Giron

Review 1:

Comments and Suggestions for Authors

Authors are commended on a well-organized paper covering a relevant topic for today's automobile industry. The paper flows well and is easy to follow, and overall, is well thought out. There are several referencing and grammatical errors, as seen in the comments concerning English language, but outside of that, there are only a few points for suggested improvements within the paper before moving forward with publication. 

In the introduction and in the conclusions, the limitations associated with costs of this technology needs to be addressed. Costs are mentioned in the last paragraph before implications, but this discussion needs to be expanded with the understanding if we move forward with the studying of this technology and further application that cost effective strategies can be addressed and implemented. Without addressing the hinderance associated with consumer trust, demand is low, and thus, costs are high. As consumer trust builds with new technology, demand will rise driving production up, and hopefully, costs come down.

Response: Thank you for your accurate and thought-proving remark., We have added a short section to the introduction.: “This aspect is of paramount importance for the AV industry, as low levels of trust translate to decreased demand, subsequently leading to higher production costs. Conversely, as consumer trust in new technology grows, demand is expected to increase, driving production levels up, and ideally, reducing costs ” (page 2 starting line 48).  In the discussion section, we have added the phrase: “that results in low demand and higher production costs” (page 10 line 450).

Another area for further exploration within the paper is focused on trust building in the form of assurance concerning data privacy. This concept is not addressed until the implications section and is only mentioned in the last item listed, number 9. This discussion needs to take place at the beginning within the introduction, and circled back to within the discussion.

Response: Thank you very much for this comment, in order to correct this we added this section to the introduction: “The integration of subjective measures related to personality and physiological traits may evoke apprehensions regarding data privacy \cite{brell_conditional_2019}. To mitigate this, it's imperative that the system guarantees full privacy and encryption in data collection. By offering complete assurance of data privacy, the system can leverage the potential benefits of data collection while fostering a balanced interaction with the AV, thereby nurturing trust and bolstering acceptance” (page 3, starting in line 114).

Further, for section 3.3.6. Body Temperature, this section needs to be expanded as the discussion seems quite simplistic where the focus is just on comfort on the passenger. The other sections discuss the value of these interactive measures to ensure alertness of the driver. There are several studies looking at temperature within the vehicle and it's influence on driving, specifically alertness of the driver. These studies would justify the need for detecting the ideal temperature for the alertness of the driver, rather than just comfort. 

Response:Thank you for this valuable suggestion. To enhance and expand the section, we have included several pertinent references and additional information in the body temperature paragraph: “Gwak et al. (2015, 2019) present evidence that changes in indoor ambient temperature can enhance both thermal comfort and occupants' arousal levels \cite{gwak_effects_2015,gwak_investigation_2019}.  Furthermore, as described by Sunagawa et al. (2023) during brief driving sessions, the individual's thermal surroundings may exert a more significant impact on the progression of drowsiness than the duration of driving \cite{sunagawa_analysis_2023}.

Variations in body temperature can be indicative of stress, anxiety, or discomfort, all of which can significantly impair cognitive functions and decision-making abilities \cite{chen_pain_2021}. In the context of autonomous vehicles, where the interaction between human and machine is nuanced and complex, recognizing and responding to such physiological cues becomes essential \cite{shin_experimental_2021}. For instance, an increase in body temperature could signal stress or anxiety, prompting the vehicle's systems to initiate calming interventions, such as adjusting the interior lighting, climate, or even suggesting a break in the journey. Furthermore, monitoring body temperature can enhance the personalization of the user experience \cite{kajiwara_evaluation_2019}. By understanding an individual's typical physiological responses under various conditions, the vehicle can preemptively adjust settings for optimal comfort and performance. This level of personalization not only improves the user experience but also fosters a deeper trust and rapport between the user and the autonomous system, crucial for the widespread acceptance and success of these vehicles. “

” (page 8, starting at line 371)

Finally, since this paper is focused on new technology and a new approach to human-vehicle interaction, references concerning previous work should be relatively new. However, over 1/4 of the references listed, around 27%, are a decade old, if not older. For those references, either remove or include newer references to support these findings. 

Thank you for your comment. To address this, the following references have been added:

Subasi, A., Saikia, A., Bagedo, K., Singh, A., & Hazarika, A. (2022). EEG-based driver fatigue detection using FAWT and multiboosting approaches. IEEE Transactions on Industrial Informatics, 18(10), 6602-6609.

Zhu, J., Zhang, Y., Ma, Y., Lv, C., & Zhang, Y. (2023, September). Designing Human-machine Collaboration Interface Through Multimodal Combination Optimization to Improve Takeover Performance in Highly Automated Driving. In 2023 IEEE 26th International Conference on Intelligent Transportation Systems (ITSC) (pp. 4895-4900). IEEE.

Chen, J., Abbod, M., & Shieh, J. S. (2021). Pain and stress detection using wearable sensors and devices—A review. Sensors, 21(4), 1030.

Kajiwara, S. (2019). Evaluation of driver status in autonomous vehicles: Using thermal infrared imaging and other physiological measurements. International Journal of Vehicle Information and Communication Systems, 4(3), 232-241.

Shin, Y., Ham, J., & Cho, H. (2021). Experimental study of thermal comfort based on driver physiological signals in cooling mode under summer conditions. Applied Sciences, 11(2), 845.

And these were removed:

Patrick, C.J.; Curtin, J.J.; Tellegen, A. Development and validation of a brief form of the 860

Multidimensional Personality Questionnaire. Psychological Assessment 2002, 14, 150–163. Place: 861 US Publisher: American Psychological Association, https://doi.org/10.1037/1040-3590.14.2.1 862

  1. 863

Kirschbaum, C.; Pirke, K.M.; Hellhammer, D.H. The ‘Trier Social Stress Test’–a tool for in- 857

vestigating psychobiological stress responses in a laboratory setting. Neuropsychobiology 1993, 858

28, 76–81. Publisher: S. Karger AG Basel, Switzerland.

Häkkänen, H.; Summala, H.; Partinen, M.; Tiihonen, M.; Silvo, J. Blink duration as an indicator 782

of driver sleepiness in professional bus drivers. Sleep 1999, 22, 798–802. https://doi.org/10.109 783

3/sleep/22.6.798.

Artaud, P.; Planque, S.; Lavergne, C.; Cara, H.; Tarriere, C.; Gueguen, B. An on-board system for 760 detecting lapses of alertness in car driving. In Proceedings of the Proceedings: International 761

Technical Conference on the Enhanced Safety of Vehicles. National Highway Traffic Safety 762

Administration, 1995, Vol. 1995, pp. 350–359.

Cellar, D.F.; Nelson, Z.C.; Yorke, C.M. The Five-Factor Model and Driving Behavior: Personality 642

and Involvement in Vehicular Accidents. Psychological Reports 2000, 86, 454–456. https: 643

//doi.org/10.2466/pr0.2000.86.2.454.

Elbanhawi, M.; Simic, M.; Jazar, R. In the Passenger Seat: Investigating Ride Comfort Measures 678

in Autonomous Cars. IEEE Intelligent transportation systems magazine 2015, 7, 4–17.

The remaining dated references represent seminal papers and findings that we consider important to retain.

Comments on the Quality of English Language

In several lines, as given below, the authors reference the authors of other papers within a sentence, but do not include the year of that publication. This is not done with every reference, and thus, to maintain consistency throughout correct the following lines so that the year of the publication is included:

240, 251, 296, 303, and 320.

For several sentences the authors are referenced with the year of the publication, however, the number associated with that reference as given in the reference list is not included. This number should be given, preferably at the end of the sentence in which this is done for most, but not all sentences where the authors are mentioned. Below are the lines where the authors are mentioned, but no reference number is given at the end of the sentence:

74, 168, 204 (25 is given, but then ? is given for the 2020 publication), 240, 267, 303, 314, 320 (no year given either), and 367

In lines 296, 362, and 396, move the number of the reference to the end of the sentence. 

In the following lines, capitalize the first word at the start of the sentence: 267, 340, 374, 439, and 441.

In lines 38, 40, and 267, remove the period before the reference number.

In line 282, add a period before EEG. 

In line 332, remove [105] at the beginning of the sentence, before the word Implementing. 

In lines 11, 60, and 464, TOR is defined, but the authors define it as Takeover Request, Take Over Requests, and Take over requests, respectively. Authors need to be consistent in how they define TOR in both the capitalization and whether they combine or separate the words take and over. Go with what is the most common within the industry and what is utilized most consistently within the research. 

Response, Comments on the Quality of English Language : Thank you immensely for providing such a highly detailed and specific review. All corrections have been diligently incorporated into the manuscript as per your guidance.

Reviewer 2 Report

Comments and Suggestions for Authors

1.      The first chapter describes the five levels of autonomous driving systems, but it does not clearly pointed out in the following sections which level of autonomous driving the proposed technology should be classified into. The application level of this technology can be analyzed in combination with the current technological.

2.      The second chapter mentions that "improving the trust and acceptance of the technology may be more important than the security measures," So do security measures mean active security or passive security? Additionally, in order to convince users of this viewpoint, stronger evidence is needed, such as putting forward some ideas about the universality and superiority of autonomous driving technology.

3.      In the general cognition, autonomous driving should be based on the active safety control strategy of the vehicle. The second chapter of the article proposes that control strategies based on human body signals can enhance trust and deliver a more natural driving experience, implicitly suggesting that trust serves safety. The authors need to clarify the relationship among current vehicle control technology, human-computer interaction and vehicle safety.

4.      Personalized interaction is the focus of this article, but security issues are always put in the first place. This article rarely mentions how to take into account security issues while conducting this personalized interaction. 

5.      The whole article consists of extensive textual descriptions but lacks sufficient visual illustrations. It is important to consider the proportion of figures to text in the article.

6.      The whole article is highly theoretical. The author can appropriately add the market research results of the autonomous driving system and the user 's experience of this kind of personalized interaction in the article.

7.      The whole article, especially the conclusion part, lacks the support of data. For example, when referring to prediction, the prediction accuracy can be appropriately added to improve the persuasiveness of the article.

Comments on the Quality of English Language

Moderate editing of English language required

Author Response

Dear Editor,

We thank the reviewers for the thought and time they put into reviewing our manuscript. And we are grateful to the reviewers for their  generous remarks and insightful comments.  We have edited the article to address their concerns and in so doing, have improved it greatly.

We believe that the article is now suitable for publication in Sensors

Dr. Jonthan Giron

Review 2:

Comments and Suggestions for Authors

  1. The first chapter describes the five levels of autonomous driving systems, but it does not clearly pointed out in the following sections which level of autonomous driving the proposed technology should be classified into. The application level of this technology can be analyzed in combination with the current technological.

Response: Thank you very much for this important clarification., In order to accommodate this, we added the sentence: “Our goal is to introduce a novel interface to aid in this aspect for AVs up to level 4, encompassing those involving TOR interaction.” (page 3 Line 113)

  1. The second chapter mentions that "improving the trust and acceptance of the technology may be more important than the security measures,"So do security measures mean active security or passive security?

Response: Improving security from the perspective of the current paper in autonomous vehicles (AVs) encompasses a multifaceted approach aimed at safeguarding passengers from potential harm or risks throughout their journey. This involves both active and passive measures to ensure their safety. Active measures involve real-time monitoring and decision-making by the AV's systems to navigate the vehicle safely through various obstacles and hazards on the road. This includes employing advanced sensor technologies such as lidar, radar, and cameras to detect and react to objects, pedestrians, and other vehicles in the vehicle's vicinity. Additionally, sophisticated algorithms are utilized to analyze the data from these sensors and make split-second decisions to avoid collisions or other dangerous situations. Passive measures focus on preemptive planning and risk mitigation strategies to minimize potential dangers before they arise. This involves extensive mapping of routes and environments, taking into account factors such as road conditions, traffic patterns, and potential hazards. By planning ahead, AVs can choose optimal paths and anticipate potential risks, thereby reducing the likelihood of accidents or incidents during the journey.

Overall, a comprehensive approach to security in AVs combines active navigation and obstacle avoidance with passive planning and risk mitigation strategies, augmented by robust cybersecurity measures to ensure the safety of passengers throughout their journey.

Additionally, in order to convince users of this viewpoint, stronger evidence is needed, such as putting forward some ideas about the universality and superiority of autonomous driving technology.

Response: In order to convince users of this point we have strengthened the point by adding the following sentence: “As described, despite significant technological advancements, current autonomous vehicles (AVs) up to level 4 still require active passenger interaction to facilitate safe traveling.” (page 1 line 38). In the introduction, our objective is to persuade readers that human-vehicle interaction remains crucial despite advancements in AV technology. We propose that by utilizing personality and psychophysiological indices to develop an advanced interface, we can enhance safety measures and improve the overall experience in terms of comfort.

  1. In the general cognition, autonomous driving should be based on the active safety control strategy of the vehicle. The second chapter of the article proposes that control strategies based on human body signals can enhance trust and deliver a more natural driving experience, implicitly suggesting that trust serves safety. The authors need to clarify the relationship among current vehicle control technology, human-computer interaction and vehicle safety.

Response: Thank you for your incisive comment, we agree with the need to provide further clarification on this aspect. To address this, we introduced a new short section to the introduction:. “ Presently, numerous sensors are being integrated into the cabins of autonomous vehicles (AVs) to facilitate advanced interaction, enabling the understanding of driver and/or passenger actions, emotions, and personal preferences. This integration aims to provide adequate functionalities and services for a safe and enjoyable drive \cite{muraliIntelligentVehicleInteraction2022}. Among these interfaces, some focus on detecting attention, emotion recognition, drowsiness, and mental workload \cite{muraliIntelligentVehicleInteraction2022}. These interfaces prioritize evaluating and ensuring the proper state of the driver/passenger to maximize safety levels during the drive using driving assistance systems \cite{muraliIntelligentVehicleInteraction2022}.” (page 2 line 71)

  1. Personalized interaction is the focus of this article, but security issues are always put in the first place. This article rarely mentions how to take into account security issues while conducting this personalized interaction.

Response: Thank you for your insightful comment. To provide further clarity, we have introduced a paragraph in the second chapter of the manuscript. “Our interaction framework involves two distinct stages, each tailored to different aspects of the autonomous vehicle (AV) experience. Firstly, our framework prioritizes safety during TOR transitions. This phase is crucial as it marks the shift from autonomous driving to human control. To ensure the safety of the drive, the AV employs comprehensive safety protocols basing the operation on real-time monitoring systems provided by the frame work. This includes rigorous testing and validation of the AV's sensor fusion algorithms and decision-making processes to accurately assess the driving environment and respond to potential hazards effectively. Additionally, clear communication channels are established to alert passengers about the transition and provide reassurance regarding the safety measures in place. Secondly, the framework seamlessly transitions into a mode focused on enhancing user comfort throughout the drive. This involves optimizing the AV's driving behavior to prioritize passenger comfort, such as smooth acceleration, braking, and steering. Furthermore, the framework incorporates personalized settings and preferences to tailor the driving experience to individual passengers, including options for climate control, entertainment, and seating adjustments. By prioritizing passenger comfort, the framework aims to enhance the overall user experience and promote trust and acceptance of autonomous driving technology. ” (page 4 line 154)

We further introduced a concluding remark that emphasizes this concept: “In summary, our interaction framework is designed to address both safety and comfort aspects of the AV journey. By effectively managing the TOR transition and prioritizing passenger comfort throughout the drive, we aim to deliver a seamless and enjoyable autonomous driving experience for all passengers.” (page 12 line 554)

  1. The whole article consists of extensive textual descriptions but lacks sufficient visual illustrations. It is important to consider the proportion of figures to text in the article.

Response: Thank you for your remark. Given that the article proposes a theoretical framework, we believe it is appropriate to include only a diagram of the suggested workflow and omit any other illustrations.

  1. The whole article is highly theoretical. The author can appropriately add the market research results of the autonomous driving system and the user 's experience of this kind of personalized interaction in the article.

Response: Thank for this comment please see response for the next remark

  1. The whole article, especially the conclusion part, lacks the support of data. For example, when referring to prediction, the prediction accuracy can be appropriately added to improve the persuasiveness of the article.

Response: Thank you for your insightful comment and the additional insights regarding the theoretical nature of the paper. As you rightly pointed out, the current manuscript outlines a theoretical framework for utilizing psychophysiology to induce implicit interaction with autonomous vehicles (AVs), alongside exploring the influence of personality traits on this interaction. Our objective is to contribute these concepts to the academic and professional communities, stimulating further exploration in the field. Subsequent to this publication, we plan to collaborate with industry partners to evaluate our hypotheses and validate our concepts in real-world settings.

Round 2

Reviewer 2 Report

Comments and Suggestions for Authors The author has modified the paper according to the suggestions. The current version meets the requirements and can be published. Comments on the Quality of English Language

 Minor editing of English language required.